# Regional Scale Impact of the COVID-19 Lockdown on Air Quality: Gaseous Pollutants in the Po Valley, Northern Italy

Giovanni Lonati *[ID] and Federico Riva [ID]

Dipartimento di Ingegneria Civile e Ambientale, Politecnico di Milano, 20133 Milano, Italy; federico3.riva@mail.polimi.it

* Correspondence: giovanni.lonati@polimi.it

**Abstract:** The impact of the reduced atmospheric emissions due to the COVID-19 lockdown on ambient air quality in the Po Valley of Northern Italy was assessed for gaseous pollutants ($NO_2$, benzene, ammonia) based on data collected at the monitoring stations distributed all over the area. Concentration data for each month of the first semester of 2020 were compared with those of the previous six years, on monthly, daily, and hourly bases, so that pre, during, and post-lockdown conditions of air quality could be separately analyzed. The results show that, as in many other areas worldwide, the Po Valley experienced better air quality during 2020 spring months for $NO_2$ and benzene. In agreement with the reductions of nitrogen oxides and benzene emissions from road traffic, estimated to be −35% compared to the regional average, the monthly mean concentration levels for 2020 showed reductions in the −40% to −35% range compared with the previous years, but with higher reductions, close to −50%, at high-volume-traffic sites in urban areas. Conversely, $NH_3$ ambient concentration levels, almost entirely due the emissions of the agricultural sector, did not show any relevant change, even at high-volume-traffic sites in urban areas. These results point out the important role of traffic emissions in $NO_2$ and benzene ambient levels in the Po Valley, and confirm that this region is a rather homogeneous air basin with urban area hot-spots, the contributions of which add up to a relatively high regional background concentration level. Additionally, the relatively slow response of the air quality levels to the sudden decrease of the emissions due to the lockdown shows that this region is characterized by a weak exchange of the air masses that favors both the build-up of atmospheric pollutants and the development of secondary formation processes. Thus, air quality control strategies should aim for structural interventions intended to reduce traffic emissions at the regional scale and not only in the largest urban areas.

**Keywords:** air quality; nitrogen oxides; emission reduction; COVID-19 lockdown; Po Valley

## 1. Introduction

Due to the outbreak of the SARS-CoV-2 (COVID-19) pandemic, in the first months of 2020 many governments and local authorities worldwide established measures to limit the spread of the virus. Depending on the country, these measures were based on a mild approach, simply relying on "social distancing" behavior for the population, or on a more severe approach (the so-called "lockdown"), additionally relying on the stoppage of commercial and productive activities, on the closure of schools and universities, and on personal mobility limitations and the obligation to remain in one's residence, for a number of consecutive weeks.

The reductions in social and economic activities and in the related atmospheric emissions caused a change in air quality that has been evidenced by satellite data [1–4]. Recently, a number of papers have been published reporting the changes in concentration levels for certain pollutants in different areas worldwide, where light or heavy restrictive measures have been implemented [5]. Most of these works refer to China [6–8], Korea [9–11], India [12], Central [13,14], and South-East Asia [15], but also to Western European countries [16–18], Africa [19,20], and North and South America [21–23]. The few studies related

to Northern Italy [24–26] usually discussed single-city case studies, typically with a time horizon limited only to the lockdown weeks.

In order to widen the perspective, both spatially and temporally, this work is focused on the air quality in the whole Po Basin, a plain area of about 45,000 km$^2$ with 25 million inhabitants in Northern Italy, during the first six months of 2020, that is, before, during and after the COVID-19 lockdown period in Italy. The Po Valley is a well-known European hot-spot for air pollution, where the compliance with air quality limits is a critical issue, especially in urban areas. Indeed, the high density of people, agriculture, and industrial and commercial activities, together with a dense network of roads and motorways, determine huge emissions, whose dispersion is limited by the presence of the mountain chains of the Alps and Apennines and by the atmospheric stability and lapse rate inversion, typical of the cold season [27–30]. Air quality remediation plans enforce progressive restrictions on old car circulation and biomass combustion for domestic heating, occasionally further strengthened under peak pollution events. However, the restrictions adopted during the late winter and spring of 2020 created an unprecedented scenario, both in terms of spatial extent and temporal continuity of the measures, especially affecting the emissions regime, and consequently, the local air quality. This work is intended to assess the impacts of these measures on air quality, focusing on gaseous pollutants, namely, nitrogen dioxide ($NO_2$), benzene, and ammonia ($NH_3$); further work will focus on particulate matter.

The Italian government and the regional authorities have implemented a series of measures to contain the spread of COVID-19 that directly or indirectly had an impact on typical life and straightforwardly had one on the emissions regime. On February 23rd, the first decree was issued with measures (including distance learning in schools and universities; limitations on the transport services of goods and people, and on local public transport; suspension of any kind of events and meetings in public or private places) to be implemented only in the municipalities where at least one positive case to the virus had been recorded. On March 1st, the same measures were enforced in 11 municipalities in Lombardy and Veneto and on March 4th in all of Italy. Additional measures were first enforced in Northern Italy on March 8th and then in the whole country on March 9th, the starting date of the national lockdown, namely, with the prohibition of any form of gathering of people in public or places open to the public. Further restrictive measures were enforced on March 11th, concerning the suspension of retail commercial activities (except for the sale of food and basic necessities), of catering services (including bars, pubs, and restaurants), and of activities relating to personal services (i.e., hairdressers, barbers, and beauticians), and finally on March 22nd, with the suspension of most industrial and commercial activities and the prohibition on people leaving their residence municipalities, other than for proven work needs, absolute urgency, and health reasons. All the restrictive measures remained in force until May 17th, when the reopening of commercial activities and intra-regional mobility was permitted; interregional travelling was allowed from June 3rd.

According to emissions inventory data of the four regions of the Po Valley (Emilia-Romagna, Lombardia, Piemonte, and Veneto), road traffic (53%) and other mobile sources (13%) emissions accounted for two-thirds of the total annual emissions of nitrogen oxides (about 320 Gg year$^{-1}$), followed by combustion processes (16% overall) and by industrial processes (15%) (Figure 1a). Total emissions of Volatile Organic Compounds (VOCs) were more distributed among the emission sectors, mainly deriving from the use of solvents, agriculture, and other sources, including nature (Figure 1b). However, as far as benzene was specifically concerned, national estimates indicated that road transport was responsible for more than 50% of its emissions in large urban areas and for an average of 47% in medium sized cities [31]. Conversely (Figure 1c), $NH_3$ was almost entirely emitted by agriculture (97%); contributions from non-industrial combustion plants, road transport, and waste treatment each accounted for 1% of the total annual emissions (about 185 Gg year$^{-1}$).

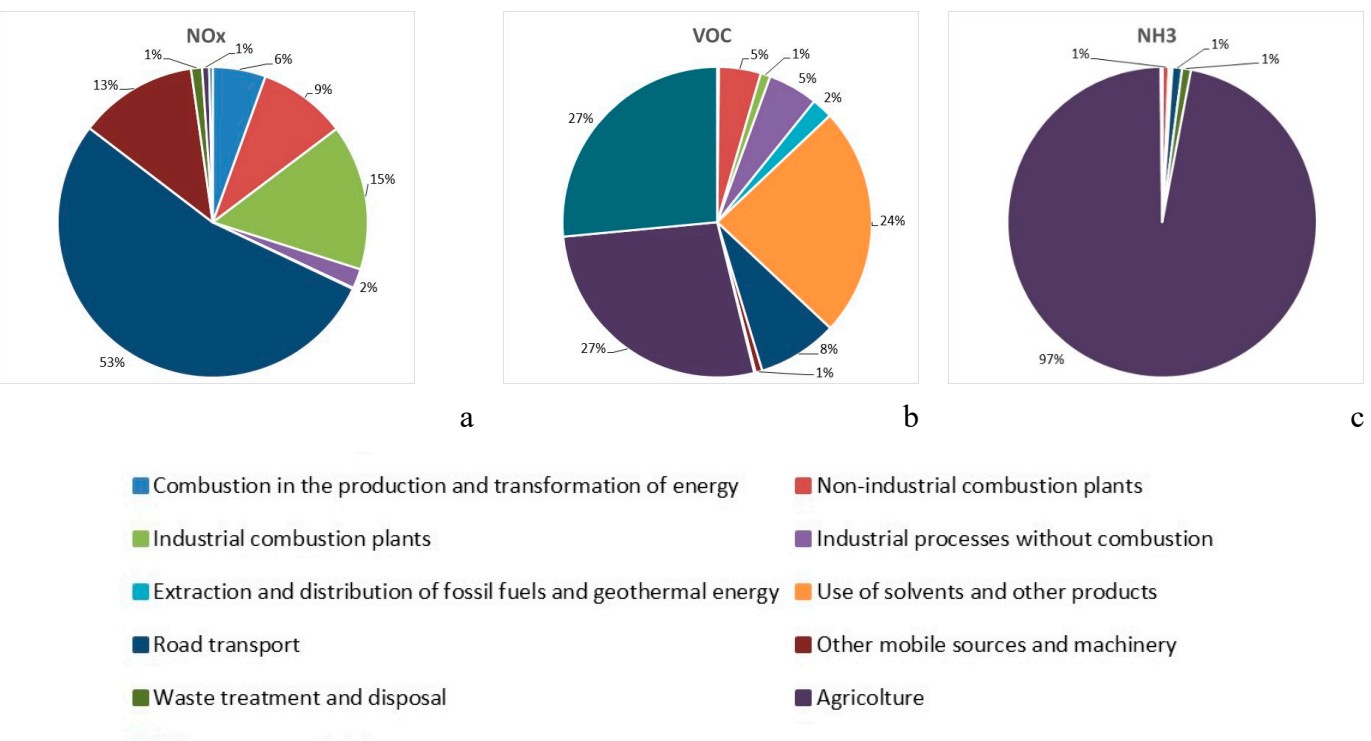

**Figure 1.** Selected Nomenclature for Air Pollution (SNAP) sectors' contributions to (**a**) NOx, (**b**) VOCs, and (**c**) $NH_3$ annual emissions in the Po Valley (elaboration of emission inventory data of Emilia-Romagna, Lombardia, Piemonte, and Veneto region).

Due to the progressive enforcement of the restrictive measures since the last week of February, atmospheric emissions varied from day to day, also depending on the pollutants in emissions, compared to their typical values (Supplementary Materials Figure S1). Estimates developed by the Environmental Agency of Lombardia [32] for NOx during the weeks from 9 March to 26 April indicated reductions around a regional average value of about −36%, but ranging between −44% (6–12 April) and −20% (9–15 March); however, emission reductions in large urban areas in April were −50%, and sometimes −60%. The main contributor to these decreases was the reduction in emissions from road traffic, which was estimated to be around −65% over the entire lockdown period, but −75% at the beginning of April. For VOCs the overall reduction was smaller, −12% over the 9 March–26 April period, because different sectors contributed to their emissions. The decrease in emissions of VOCs from road traffic was estimated in the order of −70%; a −35% reduction of benzene emission could be expected. Conversely, for $NH_3$, originating primarily from agriculture, namely, from manure management and applications of fertilizers [33], the overall reduction was practically null (−1%), even though for the road traffic contribution a −74% reduction was estimated over the reference period, but some weekly values were also around −80%.

In order to assess the impact on air quality of these emission reductions, concentration data collected all over the Po Valley area in 2020 and in the six previous years have been processed and compared on monthly, daily, and hourly bases. Data processing considered the January–June period so that pre, during, and post-lockdown conditions of air quality could be separately analyzed. Data comparisons were performed for each month individually to avoid the seasonal effects on the ambient concentration levels due to both the different emission patterns and the typical meteorological differences between winter and spring months. Data processing was performed on raw concentration values, without any adjustment for meteorological conditions, because the regional meteorological services did not report the occurrence of peculiar conditions (strong wind or heavy rainfall events),

able to alter either the long-term concentrations (i.e., monthly values) or their temporal patterns on a daily basis. Additionally, according to analyses performed on meteorological variables in other studies, the average meteorological conditions during the 2020 lockdown period did not differ from those of the previous years [34,35]. Thus, the observed variations in the concentration levels can be ascribed to the change in human activity. This work intended to assess the impact of human activity on air quality over the whole Po Valley in general and its different geographical areas in particular, while providing specific insights for the urban areas, where the compliance with air quality limits is more critical.

## 2. Material and Methods

### 2.1. Air Quality Monitoring Networks

In the Po Valley, air quality monitoring networks are separately operated by the Regional Environmental Agencies (ARPA) of Emilia-Romagna, Lombardia, Piemonte, and Veneto region. According to the European Union Directive 2008/50 [36], concentration data for reference atmospheric pollutants are routinely collected at fixed monitoring sites distributed all over the territory (Figure 2). As monitoring does not consider all the pollutants at all sites, the number of stations available is pollutant-dependent. The numbers of monitoring stations in operation during the last seven years (2014–2020) are summarized in Table 1. $NO_2$ data are available with hourly time resolution, whereas benzene and ammonia ($NH_3$) are available with daily resolution; all concentration values are expressed in μg m$^{-3}$ and refer to 20 °C and 101.3 kPa.

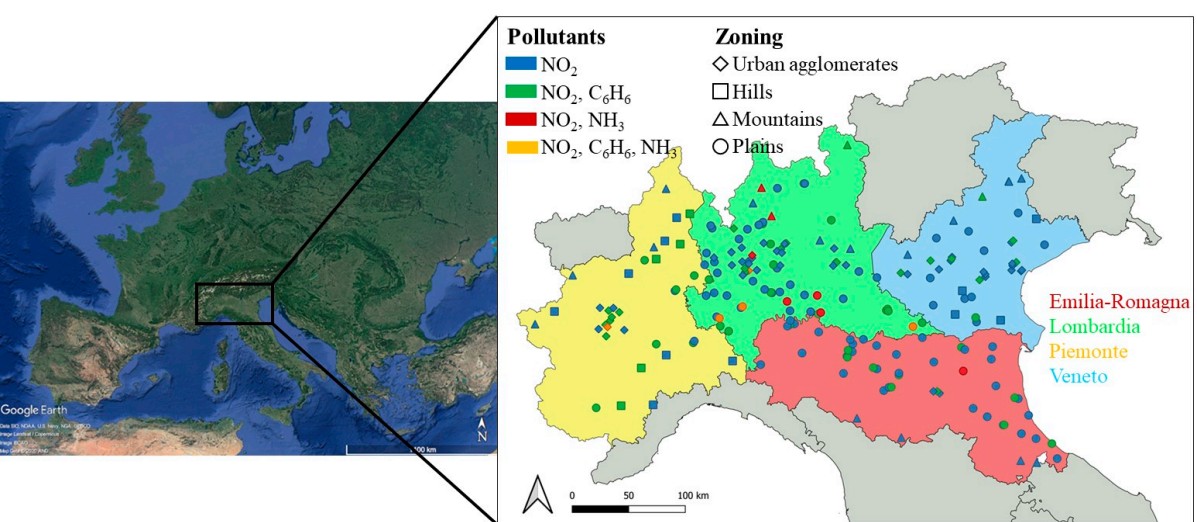

**Figure 2.** Spatial distribution of the air quality monitoring stations in operation during the last seven years (2014–2020) in the regions of the Po Valley. (Satellite image source: Google Earth; marker color: monitored pollutants; marker shape: station classification by location).

**Table 1.** Number of monitoring stations in operation during the last seven years (2014–2020) in the regions of the Po Valley.

| Region | $NO_2$ | Benzene | $NH_3$ |
|---|---|---|---|
| Emilia-Romagna | 43 | 9 | - |
| Lombardia | 82 | 22 | 10 |
| Piemonte | 52 | 22 | 4 |
| Veneto | 41 | 9 | - |
| Total | 218 | 62 | 14 |

Monitoring stations were classified according to two different criteria based on geographical location and on exposure to emission sources. The former makes distinctions among the stations located in parts of the territory ("zones") with given features (e.g., plains areas, and mountain areas) and those located in urban agglomerations (cities and conurbations of cities with more than 250,000 inhabitants); the latter makes distinctions using the context of where the stations are located (urban, suburban, industrial, or rural), with additional details for the types of emission sources they are exposed to (i.e., traffic or background, which is influenced by the integrated contribution of all upwind sources).

As the land zoning for station classification is performed at the regional level, we had a total of 23 different classes, often with common features (i.e., 10 for urban agglomerations), but in other cases reflecting the particular and specific features of the territory of the Po Valley. However, based on both the geographical features of the monitoring sites and on their typical concentration levels, we were able to reduce the number of zones down to four, namely: urban agglomerations, plains, hills, and mountains. The urban agglomerations zone encompasses the stations located in the large conurbations of Lombardia and Piemonte (cities of Milano, Brescia, Bergamo, and Torino) and in the largest cities of Veneto and Emilia-Romagna (Padova, Treviso, Venezia, Verona, Vicenza, and Bologna); the plains zone encompasses all the stations in the flat area of the Po Valley, including those located in the small provincial capitals and in the valley floor of Northern Lombardia; the hills zone encompasses the stations in the hilly areas of Piemonte and Veneto; the mountains zone encompasses the stations located in prealpine and alpine areas and in the Apennines of the Emilia-Romagna. The combination of the two classification criteria (zoning and type) allowed the creation of stratified datasets out of the overall dataset, useful for detailed investigations and comparisons of the effects of lockdown on air quality, for instance, through separately analyzing data from traffic stations and background stations in urban agglomerations.

*2.2. Air Quality Data Processing*

Air quality data were accessed and downloaded through the institutional websites of the Regional Environmental Agencies and organized in a single dataset for the 2014–2020 period. In order to assess the effect of the reduced atmospheric emissions in the Po Valley as a consequence of the COVID-19 lockdown, data for the first semester of 2020 were compared with the mean of the six previous years (2014–2019). Analyses were developed with different time resolutions: first, with respect to the monthly mean concentrations, and then with respect to the time pattern of the daily mean concentrations at monthly and weekly levels; for $NO_2$, due to the 1-hour time resolution of the data, the time pattern of the hourly mean concentration at the daily level was investigated too. In order to account for the seasonality of ambient concentrations, comparisons were separately performed for each single month, considering both the whole dataset and stratified datasets according to the station classification criteria, assessing absolute and relative variations in concentration levels. Statistical tests were applied to aggregate indicators (i.e., overall monthly mean values, $\alpha = 0.95$), and for the whole monthly data distributions (Kolmogorov-Smirnov test, $\alpha = 0.99$) of 2020 and 2014–2019 datasets. For time pattern analyses at the monthly level, data distributions for each single calendar day were assembled, assessing the frequency of occurrence of 2020 data within corresponding concentration ranges observed in the previous six years. These ranges were defined based on the $q1_{14-19}$, $q2_{14-19}$, and $q3_{14-19}$ values, respectively representing the 1st, 2nd, and 3rd quartiles computed for the 2014–2019 daily datasets. The interquartile range $IQR_{14-19}$ (i.e., the concentration interval between the third and the first quartile, $q3_{14-19}$–$q1_{14-19}$) was also used for the assessment. At weekly and daily levels, the time patterns of the average week and day in 2020 and 2014–2019 were graphically compared.

## 3. Results and Discussion

### 3.1. NO$_2$

The panels of Figure 3, where the monthly mean concentrations for 2020 are plotted against the corresponding values for 2014–2019, clearly show the typical seasonality of NO$_2$ concentration levels and the impact of COVID-19 lockdown on air quality in the Po Valley. Note that during the cold season, the regional mean and maximum values are typically about twice as high as during the warm season. In January, the two datasets displayed rather similar features, with statistically non-significant differences for both the overall means (42.5 vs. 42.1 µg m$^{-3}$) and for the whole data distribution (Table 2). Relative variations of the monthly means were almost equally split between positive (increase) and negative (decrease) values, but with 50% of the stations being in the ±10% range. Conversely, from February on, together with the usual seasonal pattern of the concentration levels, we can see the progressive shift of the 2020 datasets towards lower values, with both significantly lower overall means and a different data distribution (Figure 4). The reductions of the overall mean ranged between −4.6 µg m$^{-3}$ (February, −12.1%) and −11.8 µg m$^{-3}$ (March, −37.4%), but in relative terms were even higher in April (−9.3 µg m$^{-3}$, −40.9%) in the full lockdown period. In February, relative reductions of the monthly mean were observed at about 80% of the stations—but mostly (87%) as little as −30% (−5.1 µg m$^{-3}$ on average); in March and April, reductions were observed at almost all the stations (Figure S2) and to a larger extent. In March, reductions were observed at 98.6% of the stations, with 28.4% being down to −30% (−5 µg m$^{-3}$) and 71.6% between −50% and −30% (−14.6 µg m$^{-3}$). Figures for April were quite similar, with reductions observed at 94% of the stations but even stronger in their relative magnitude: 25.6% down to −30% (−2.7 µg m$^{-3}$); and 74.1% were between −50% and −30% (−12.3 µg m$^{-3}$). In spite of the progressive loosening of the lockdown measures, reductions were still largely present in May (94%) and June (91.2%), but to a decreasing relative extent: in May, reductions by 30% (−2.8 µg m$^{-3}$) were 38.9% and those in the 30% to 50% range (−10.6 µg m$^{-3}$) were 61.1%, whereas, conversely, in June they were 61.1% (−3.2 µg m$^{-3}$) and 38.9% (−8.6 µg m$^{-3}$).

**Table 2.** Summary statistics for the NO$_2$ monthly mean concentrations (µg m$^{-3}$) in 2020 and 2014–2019 (q1, q2, and q3: 1st, 2nd, and 3rd quartiles; p5 and p95: 5th and 95th percentiles; N = number of observations; K-S: Kolmogorov-Smirnov test).

| Parameter | January | | February | | March | | April | | May | | June | |
|---|---|---|---|---|---|---|---|---|---|---|---|---|
| | 2020 | 2014–2019 | 2020 | 2014–2019 | 2020 | 2014–2019 | 2020 | 2014–2019 | 2020 | 2014–2019 | 2020 | 2014–2019 |
| Mean | 42.5 | 42.2 | 33.6 | 38.3 | 19.7 | 31.5 | 13.5 | 22.8 | 12.5 | 19.3 | 13.6 | 18.3 |
| St. dev. | 15.5 | 16.0 | 13.2 | 15.8 | 8.3 | 14.4 | 6.2 | 11.8 | 5.8 | 11.3 | 6.7 | 10.5 |
| Minimum | 2.1 | 1.2 | 1.4 | 1.9 | 2.4 | 1.5 | 1.1 | 1.6 | 0.7 | 0.7 | 0.2 | 1.0 |
| Maximum | 88.4 | 113.1 | 80.2 | 99.3 | 47.3 | 97.3 | 36.4 | 76.2 | 30.5 | 80.9 | 38.5 | 64.8 |
| q1 | 33.9 | 32.3 | 26.5 | 28.4 | 14.4 | 22.2 | 8.9 | 14.7 | 8.5 | 11.6 | 9.1 | 10.9 |
| q2 | 41.7 | 40.9 | 33.7 | 36.7 | 19.6 | 30.0 | 13.1 | 20.6 | 12.1 | 16.6 | 12.7 | 16.1 |
| q3 | 50.7 | 51.3 | 41.5 | 47.7 | 24.3 | 40.3 | 17.2 | 29.2 | 15.3 | 24.7 | 16.9 | 23.2 |
| p5 | 15.3 | 17.2 | 11.3 | 14.0 | 6.6 | 9.5 | 4.3 | 7.0 | 4.8 | 5.5 | 5.2 | 5.5 |
| p95 | 71.2 | 70.3 | 56.3 | 66.2 | 34.8 | 56.6 | 23.2 | 44.8 | 24.4 | 40.7 | 26.9 | 38.6 |
| N | 218 | 1275 | 218 | 1278 | 218 | 1277 | 218 | 1276 | 218 | 1278 | 218 | 1283 |
| Means Test | non reject | | reject | | reject | | reject | | reject | | reject | |
| K-S test | non reject | | reject | | reject | | reject | | reject | | reject | |

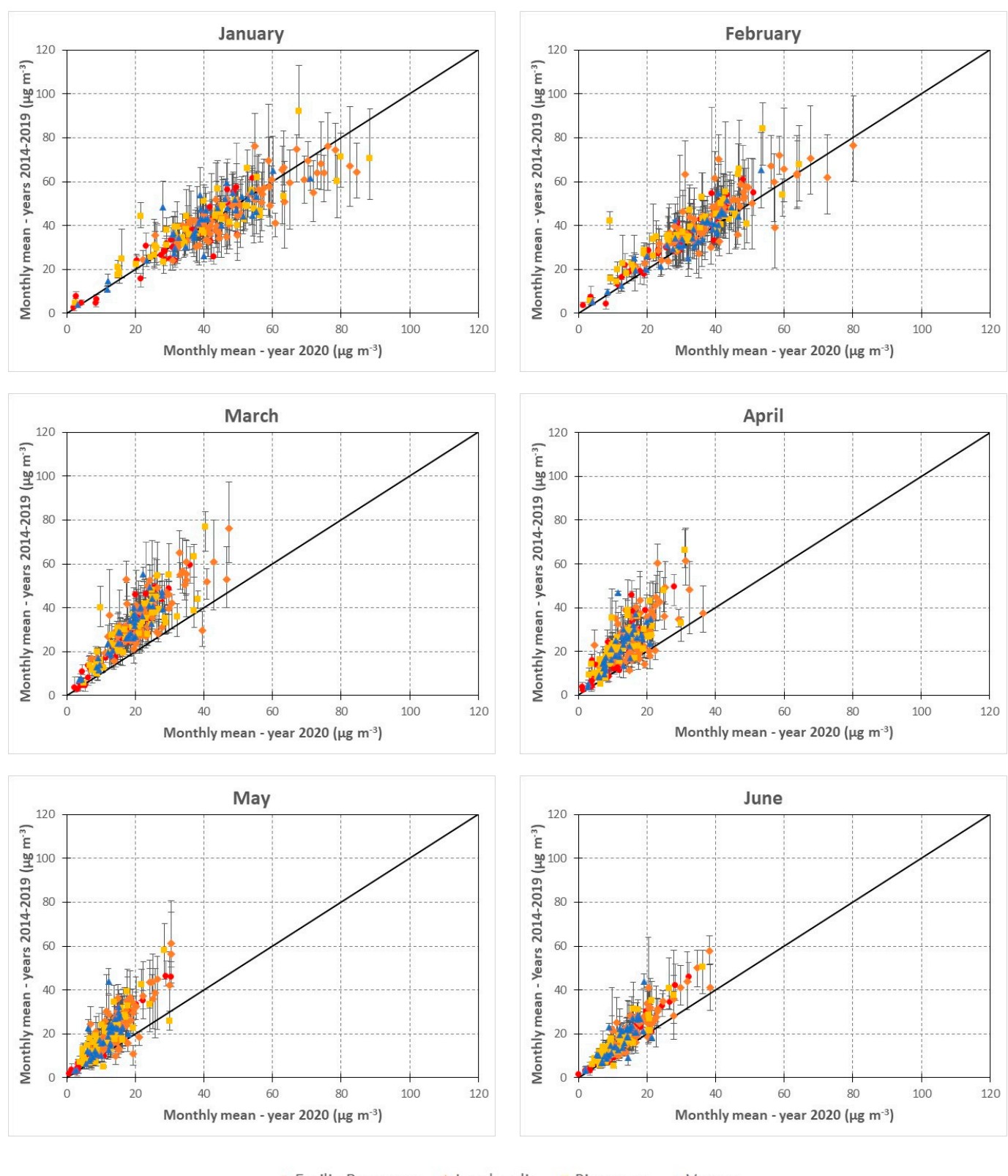

**Figure 3.** Scatterplots of the monthly mean NO$_2$ concentrations for 2020 versus the corresponding values for 2014–2019. Error bars show the minimum–maximum range for 2014–2019.

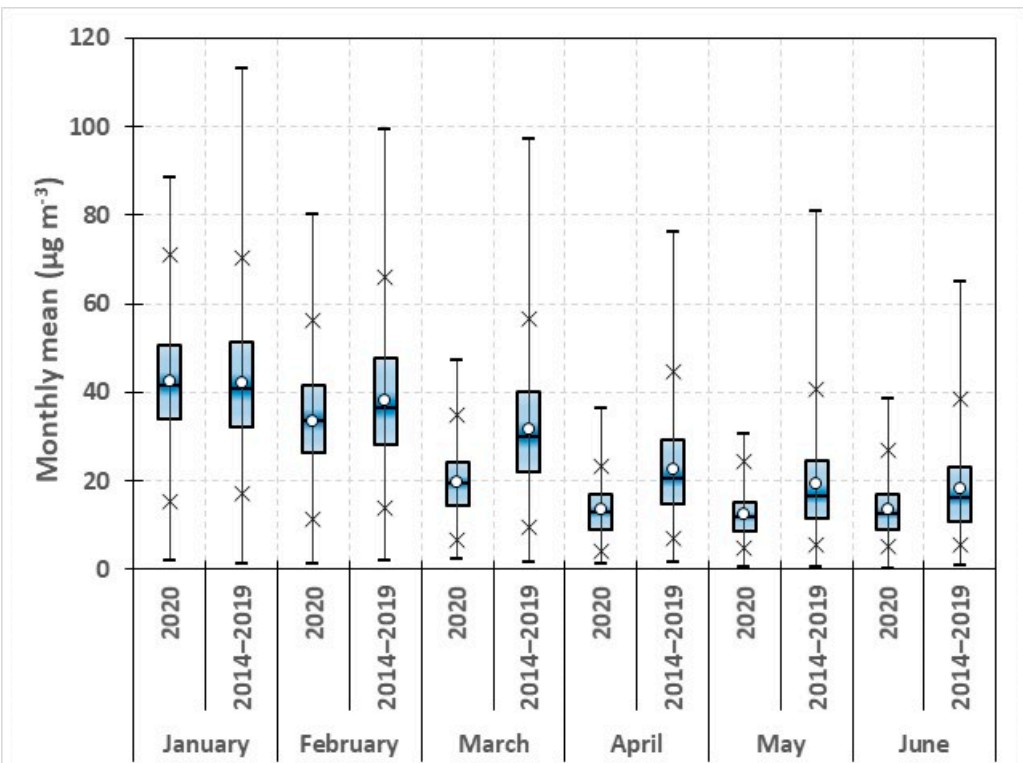

**Figure 4.** Box and whisker plots for the distributions of the monthly mean NO$_2$ concentrations in 2020 and 2014–2019. (whiskers: minimum–maximum range; crosses: 5th–95th percentile range; box: q3–q1 range).

The analysis of the stratified dataset by zones shows that the different areas of the Po Valley were rather homogeneously affected by the emission reductions due to lockdown. Both the time pattern of the relative reductions, U-shaped with the minimum values in April, and their magnitude, in particular as mean values, were fairly common to the different areas (Figure 5). However, a slightly higher average relative reduction (−42%, −14.6 µg m$^{-3}$) and larger maximum relative reductions (−72%, −35.1 µg m$^{-3}$) were observed for the stations of the urban agglomerates in the March–May period compared with the other locations—all about −36% (−7.4 µg m$^{-3}$) and with maximum values around −67% (−27.3 µg m$^{-3}$), respectively. The further stratification of the urban agglomerates subset by station type showed higher relative reductions at traffic-exposed sites (−47% on the average, ranging between −75% and −25%) than at urban background sites (−39% on the average, ranging between −72% and −3%); in absolute terms, at traffic stations the reduction (−20.3 µg m$^{-3}$) almost doubled that of the background stations (−10.8 µg m$^{-3}$). More generally, the stratification by station type highlighted that traffic sites experienced larger reductions (−41% on the average, −15.5 µg m$^{-3}$) than background sites (−32%, −6.6 µg m$^{-3}$), particularly rural background sites (−19%, −2.6 µg m$^{-3}$). Interestingly, data from one traffic station located near to the Milano-Torino motorway reported relative reductions of around −70% (about −30 µg m$^{-3}$) during the lockdown period and only around −30% (−10 µg m$^{-3}$) from May on. The extent of the concentration reduction observed during the lockdown was consistent with the emission reduction on motorways, estimated to be in the order of −70% for passenger cars and light duty vehicles and −50% for heavy duty vehicles [32].

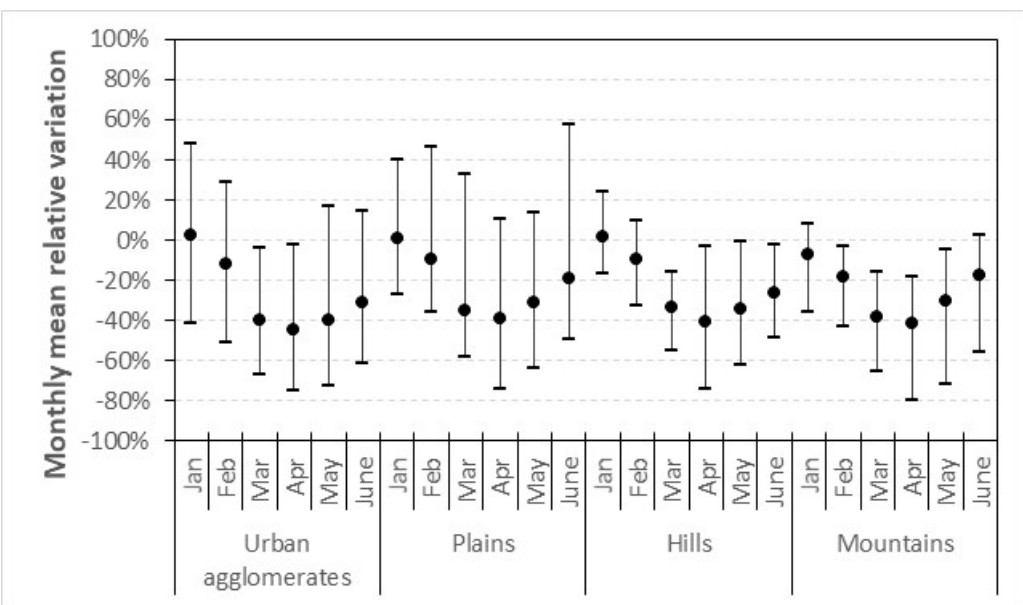

**Figure 5.** Time pattern of the relative variation of NO$_2$ monthly concentrations in 2020 with respect to 2014–2019 stratified by station zone (black dot: mean value; bars: minimum–maximum range).

The time patterns of the overall daily mean and median concentrations in 2020 calendar days are compared with the corresponding values of 2014–2019 period in Figure 6, where concentration ranges with respect to the quartiles of the 2014–2019 distributions are also plotted. The progressive reduction of the concentration values over the whole Po Valley is clearly evident and well summarized by Table 3. In January and February, the overall mean concentrations were in line with the values observed in the 2014–2019 period, almost always falling within the interquartile range (IQR$_{14–19}$ = q3$_{14–19}$–q1$_{14–19}$, roughly about 30–50 $\mu$g m$^{-3}$) and the maximum values, in spite of some peaks (e.g., 2nd week of January, February 8th–10th) and sinks (e.g., first week of February) driven by the peculiar meteorological conditions in those days of 2020. However, a certain decrease in concentration levels appeared at the end of February, when the lockdown was already enforced in small municipalities in the middle of the Po Valley. Conversely, in March and April, they were always below q2$_{14–19}$ (<30 $\mu$g m$^{-3}$), and even below q1$_{14–19}$ (<15 $\mu$g m$^{-3}$) in 50% of the days, with the differences between mean values usually being in the 5−10 $\mu$g m$^{-3}$ range, but occasionally as high as 12–15 $\mu$g m$^{-3}$. Maximum daily values of 2020 were practically always lower than in the past, lying in the q3$_{14–19}$–max$_{14–19}$ range, and even within the IQR$_{14–19}$ sometimes, with values getting closer and closer to q3$_{14–19}$, down to around 30–40 $\mu$g m$^{-3}$, in late April. In May, and most of all, in June, concentrations progressively rose, but still mainly remained below the median values of the previous years (96.7% in May, 86.7% in June). Indeed, except for the first days of May, when lockdown measures were still in force, the differences between daily mean values became smaller and smaller, mostly in the 3–6 $\mu$g m$^{-3}$ range in May and in the 2–5 $\mu$g m$^{-3}$ range in June. Nevertheless, the pattern of the maximum values showed a weak growing trend, with values up to 40–50 $\mu$g m$^{-3}$ in the end of June, that is, about 20 $\mu$g m$^{-3}$ less than in the previous years.

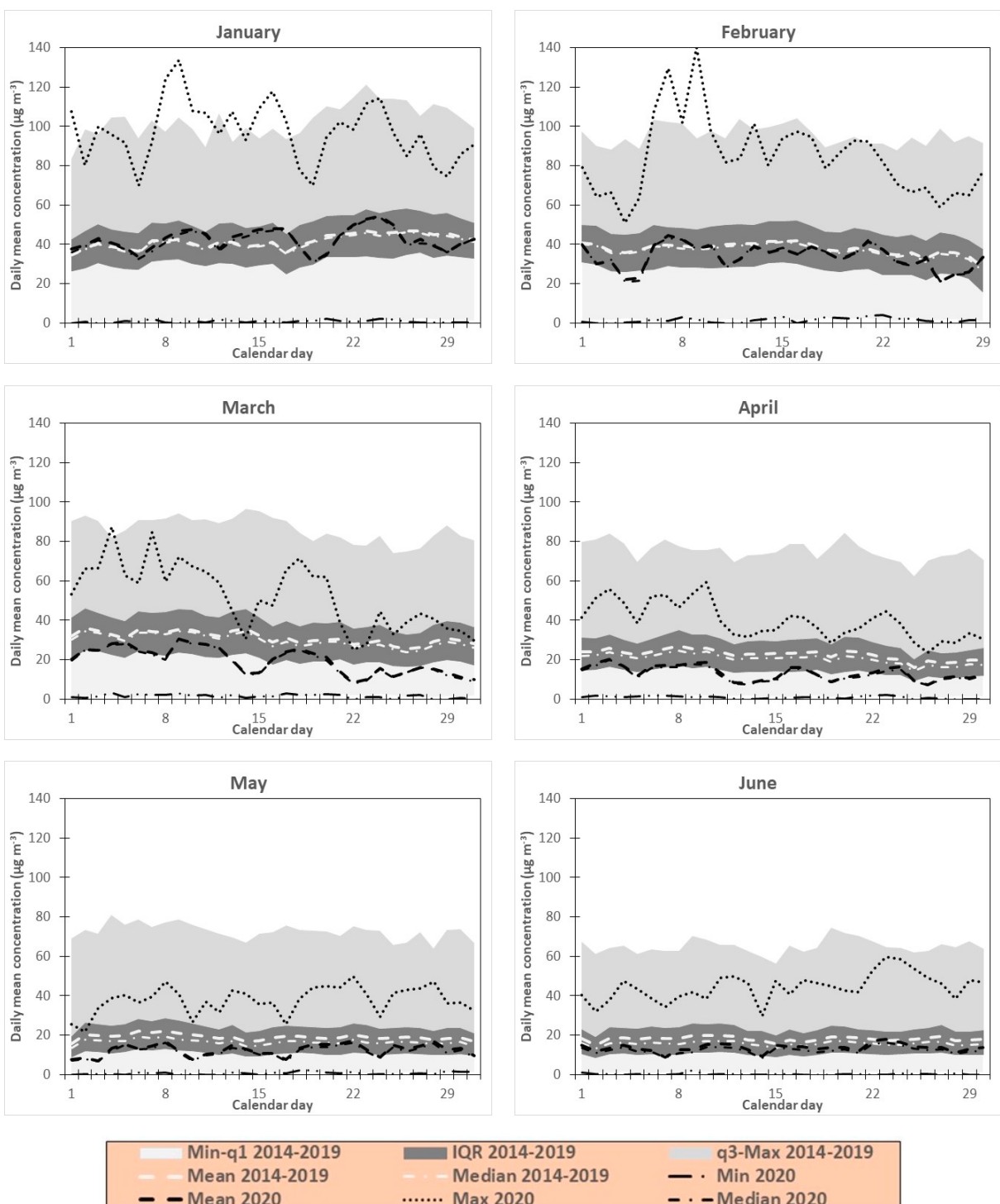

**Figure 6.** Time pattern of NO$_2$ overall daily mean concentrations for 2020 versus the corresponding values for 2014–2019. Shaded areas show concentration ranges for 2014–2019.

**Table 3.** Frequency distribution (fractional number of days in each month) of the 2020 overall daily mean $NO_2$ concentrations by concentration range observed for the 2014–2019 period ($q1_{14-19}$, $q2_{14-19}$, and $q3_{14-19}$: 1st, 2nd, and 3rd quartiles).

| Range | January | February | March | April | May | June |
|---|---|---|---|---|---|---|
| $<q1_{14-19}$ | 0.0% | 17.2% | 51.6% | 50.0% | 29.0% | 6.7% |
| $q2_{14-19}-q1_{14-19}$ | 32.3% | 48.3% | 48.4% | 50.0% | 67.7% | 80.0% |
| $q3_{14-19}-q2_{14-19}$ | 64.5% | 34.5% | 0.0% | 0.0% | 3.2% | 13.3% |
| $>q3_{14-19}$ | 3.2% | 0.0% | 0.0% | 0.0% | 0.0% | 0.0% |

The stratified analyses of the time patterns give further evidence to the piece of information obtained from monthly averaged data, namely, as far as the difference between traffic and background stations is concerned. Larger concentration reductions were consistently observed at traffic stations during the whole lockdown period, namely, in April (Figure 7). Daily means were at least 10 µg m$^{-3}$ lower than the average of the previous years—up to 20 µg m$^{-3}$ lower in some periods; in these periods maximum values were 50–60 µg m$^{-3}$ lower than their 2014–2019 means (Figure 7 top left panel). At background stations the reductions were less relevant and had reduced fluctuations, regarding both daily mean and maximum values: daily means were usually 3–8 µg m$^{-3}$ lower than in 2014–2019, but up to 10 µg m$^{-3}$ on the central days of the month, as for the traffic stations; daily maximum values showed an appreciable decrease only in the second half of the month, when they were about 20 µg m$^{-3}$ lower than the average of the previous years (Figure 7 top right panel). Focusing the analysis on urban agglomerations, the effect of the lockdown on traffic emissions was further evidenced: at traffic stations, the daily concentrations usually showed reductions in the 15–20 µg m$^{-3}$ range, but also as high as 30 µg m$^{-3}$; at background stations, reductions were most frequently in the 5–15 µg m$^{-3}$ range only (Figure 7 bottom panels). Interestingly, at the traffic stations of the largest urban agglomerates (cities of Milano and Torino) the difference between 2020 and the previous years was even larger, systematically in the order of 20–30 µg m$^{-3}$, showing the dominating role of traffic emissions in urban traffic hotspots. However, the role of traffic emissions in $NO_2$ ambient levels was even more highlighted by the reductions observed at the mentioned Milano-Torino motorway station, where daily means in April were 20–30 µg m$^{-3}$ lower than in the reference period for 12 days but up to 30–40 µg m$^{-3}$ lower for 18 days.

Regardless for the zone and station type, the 2020 time patterns displayed a rather clear 7-day cycle, corresponding to the weekly cycle, with the lowest concentrations on Sundays (e.g., March 8th and 15th or April 5th and 12th). However, such a temporal scheme was more evident at traffic stations, which are more directly influenced by traffic emissions, than at background stations, and specifically, at rural background stations (Figure 7). Indeed, even though strongly reduced during the lockdown period, road traffic was not completely gone, as alimentary goods delivery was active and personal travel for some workers (e.g., sanitary operators, alimentary markets staff, and post deliveries) was allowed. In the panels of Figures 6 and 7, 2020 Sundays' data are compared with the working days' average of the previous years; the weekly cycle was specifically investigated in order to have proper day-of-the-week comparisons. These analyses confirmed the systematically lower values throughout the lockdown and post-lockdown weeks (Figure S3), and the reduction of concentration levels on Sundays. On average, Sundays' concentration was 6.0 µg m$^{-3}$ (−28.4%) lower than on weekdays in March, 5.1 µg m$^{-3}$ (−35.9%) in April, and 5.6 µg m$^{-3}$ (−40.3%) in May; corresponding figures for the 2014–2019 period are 7.3 µg m$^{-3}$ (−22.4%) in March, 6.7 µg m$^{-3}$ (−28.1%) in April, and 5.8 µg m$^{-3}$ (−29.1%) in May, all larger in absolute terms, but they are 6% to 11% smaller in relative terms given the normally higher ambient levels. For the spring period, Sundays' concentrations during lockdown could be regarded as an indicative value of the $NO_2$ background level in the Po Valley, in the order of about 10 µg m$^{-3}$ in urban agglomerates and 6–7 µg m$^{-3}$ at rural sites.

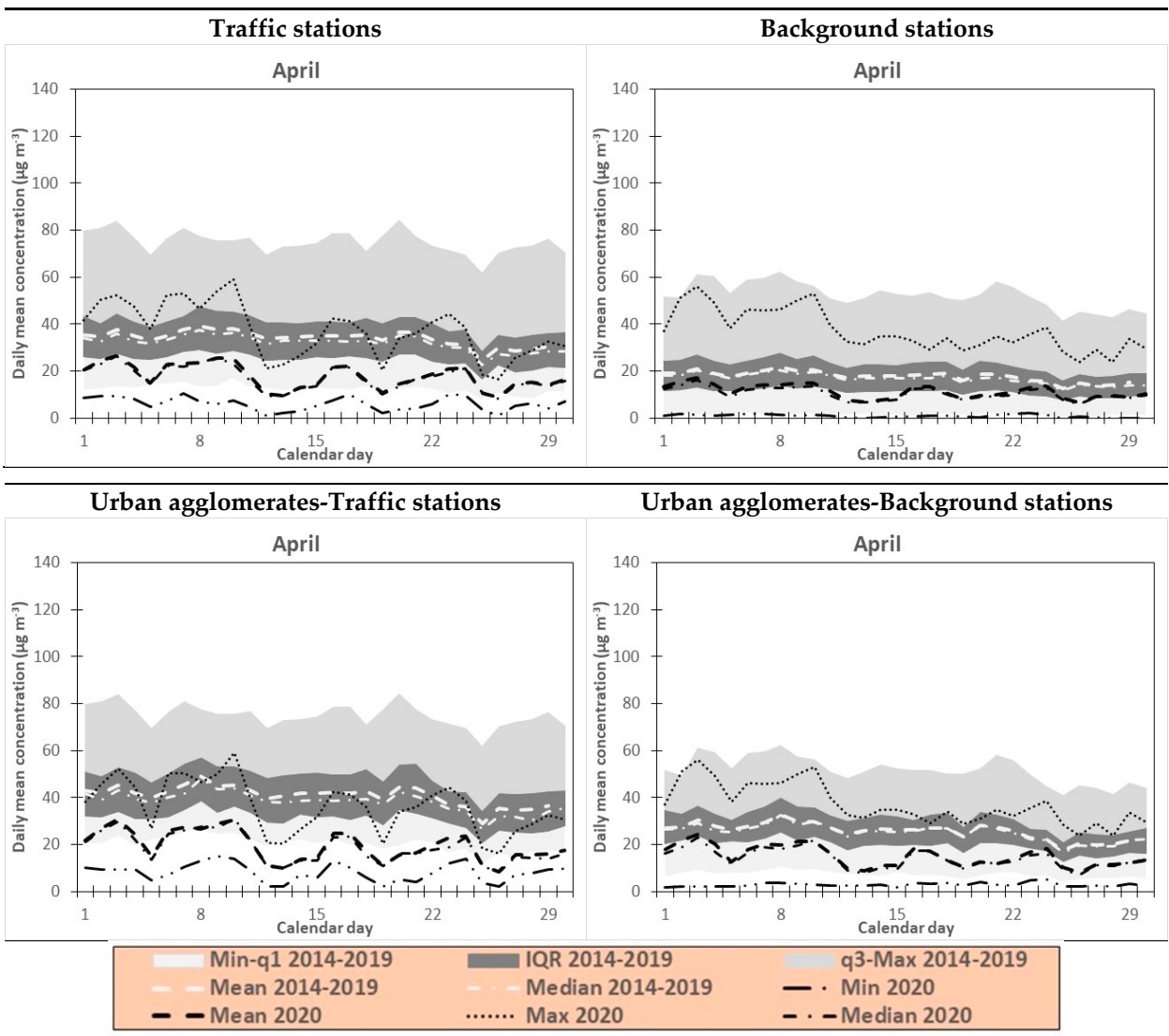

**Figure 7.** Time pattern of NO$_2$ daily mean concentrations for 2020 versus the corresponding values for 2014–2019. Shaded areas show concentration ranges for 2014–2019. Top panels: traffic stations (left) and background stations (right). Bottom panels: traffic stations in urban agglomerates (left) and background stations in urban agglomerates (right).

Finally, inspection of the daily patterns on an hourly basis showed that during lockdown, traffic emissions, even though strongly reduced, were still present, as suggested by the typical rush-hour concentration peaks in the morning and in the evening (Figure S4). Actually, the concentration levels were almost uniformly reduced during the whole day, in the order of 8–10 μg m$^{-3}$ in March and April (−36% and −40%, respectively), and 4–6 μg m$^{-3}$ (−33%) in May. However, the peak during the evening rush hours was more reduced and progressively smoothed, especially in April and May: during these hours, reductions as high as 15–18 μg m$^{-3}$ (−41%) in March, 11–14 μg m$^{-3}$ (−51%) in April, and 7–9 μg m$^{-3}$ (−39%) in May were observed. Consistently with their direct exposure to traffic emissions, traffic stations showed larger reductions in both morning and evening rush hours: in urban agglomerations, namely, reductions in the order of −55% to −35% in the morning and −60% to −50% in the evening, exceeded those of the background stations, which were only in the order of −40% to −30% in the morning and −50% to −40% in the evening.

All these results show that the measures enforced during the lockdown period had a significant and prolonged impact on NO$_2$ concentration levels, resulting in their generalized reduction all over the Po Valley. However, the magnitude of the observed reductions was strongly related to the geographical location and the exposure of the monitoring

station to the emission sources, especially road traffic. Indeed, as the reduction of the regional background was about −20%, urban areas had reductions in the order of −50% to −40%. These numbers are consistent with estimates reported for Milan metropolitan area only [25], with the reduction of tropospheric $NO_2$ in Milan (−47 ± 15%) estimated by columnar data [37], and with data from other European cities such as Barcelona (−51.4% to −47.0%) [18]. The observed reductions were the consequence of the reduced traffic flows all over the region, with emission reductions estimated to be in the order of −35% compared to the regional average, but these were even higher at urban traffic sites during the full lockdown period. The substantial agreement between the emission reductions and the generalized air quality improvement confirms the reliability of the inventory data used to assess the emissions scenario for the lockdown period. This agreement also confirms the source apportionment results for $NO_2$ in large urban areas [38], where concentration levels are mostly determined by very local and urban traffic emissions, but with an important contribution from regional background too.

*3.2. Benzene*

In general, the major findings resulting from the analyses of the benzene datasets are similar to those of $NO_2$, as far as the monthly mean variations and their time patterns are concerned and regarding the different extents to which stations have been affected by the emission reductions. Ambient concentrations in January 2020 were in agreement with the levels recorded in 2014–2019, with similar distributions and the same overall mean of 2.4 μg m$^{-3}$ (Table 4). Positive relative deviations of the monthly means slightly prevailed over negative variations (55% vs. 45%), with 39% of the stations being in the ±10% range. From February on, we can see the progressive shift of the 2020 datasets towards lower concentration levels, with both significantly lower overall means and different data distributions, namely, as far as maximum values and high percentiles are concerned (Figure 8 and, Figure S5). The reductions of the overall mean were roughly −0.4 μg m$^{-3}$ in February and March (−25.4% and −33.7%, respectively), −0.25 μg m$^{-3}$ in April and May (−35.1% and −43.3%, respectively), and −0.15 μg m$^{-3}$ in June (−33.9%). Since February, relative reductions of the monthly mean were observed at about 90% of the stations (Figure S6); however, the extent of these reductions was greater in March, April and May—when reductions in the −50% to −30% range were observed at 70% of the stations (−0.4 μg m$^{-3}$ on average, −1.1 to −0.1 μg m$^{-3}$ range)—than in February (40%) and June (50%). The outcome of the stratified dataset analysis for benzene confirmed that the Po Valley was rather homogeneously affected by the emission reduction due to the March–May lockdown, again with slightly higher average relative reductions (−40%, −0.40 μg m$^{-3}$) in the urban agglomerates than at the stations of the plains and hills sites (−30%, −0.24 μg m$^{-3}$); smaller and fairly constant reductions, in the order of −20%, were observed at mountain stations during the whole period (Figure 9). The impact of traffic emissions on benzene ambient levels was highlighted by the larger reductions generally observed at traffic stations (−41% on the average, −0.34 μg m$^{-3}$) than at background stations (−32%, −0.20 μg m$^{-3}$), especially when the urban agglomerates were concerned: traffic stations showed lockdown reductions of around −45% (−0.48 μg m$^{-3}$), and background stations, around −35% (−0.31 μg m$^{-3}$). The peculiar features of the monitoring station located near to the Milano-Torino motorway emerged from benzene data too; however, the difference with respect to the other traffic stations was less marked than for $NO_2$.

**Table 4.** Summary statistics for the benzene monthly mean concentrations ($\mu$g m$^{-3}$) in 2020 and 2014–2019 (q1, q2, and q3: 1st, 2nd, and 3rd quartile; p5, p95: 5th and 95th percentiles; N = number of observations; K-S: Kolmogorov-Smirnov test).

| Parameter | January | | February | | March | | April | | May | | June | |
|---|---|---|---|---|---|---|---|---|---|---|---|---|
| | **2020** | **2014–19** | **2020** | **2014–19** | **2020** | **2014–19** | **2020** | **2014–19** | **2020** | **2014–19** | **2020** | **2014–19** |
| Mean | 2.4 | 2.4 | 1.4 | 1.8 | 0.8 | 1.2 | 0.4 | 0.7 | 0.3 | 0.5 | 0.3 | 0.4 |
| St. dev. | 1.0 | 0.9 | 0.5 | 0.7 | 0.3 | 0.5 | 0.2 | 0.3 | 0.1 | 0.4 | 0.2 | 0.3 |
| Minimum | 0.6 | 0.3 | 0.3 | 0.1 | 0.3 | 0.2 | 0.1 | 0.0 | 0.1 | 0.0 | 0.0 | 0.0 |
| Maximum | 5.5 | 5.2 | 2.8 | 5.1 | 1.4 | 3.0 | 0.9 | 3.2 | 0.7 | 3.2 | 1.0 | 3.9 |
| q1 | 1.7 | 1.8 | 0.9 | 1.3 | 0.5 | 0.8 | 0.3 | 0.5 | 0.2 | 0.3 | 0.2 | 0.3 |
| q2 | 2.4 | 2.3 | 1.4 | 1.8 | 0.8 | 1.2 | 0.4 | 0.6 | 0.3 | 0.5 | 0.3 | 0.4 |
| q3 | 2.8 | 2.9 | 1.7 | 2.3 | 1.0 | 1.4 | 0.5 | 0.8 | 0.3 | 0.6 | 0.3 | 0.5 |
| p5 | 1.2 | 0.9 | 0.7 | 0.8 | 0.4 | 0.4 | 0.2 | 0.2 | 0.1 | 0.1 | 0.1 | 0.1 |
| p95 | 3.9 | 4.0 | 2.3 | 3.1 | 1.3 | 2.1 | 0.7 | 1.3 | 0.5 | 1.2 | 0.6 | 1.0 |
| N | 62 | 353 | 62 | 355 | 62 | 355 | 62 | 357 | 62 | 357 | 62 | 356 |
| Means Test | non reject | | reject | | reject | | reject | | reject | | reject | |
| K-S test | non reject | | reject | | reject | | reject | | reject | | reject | |

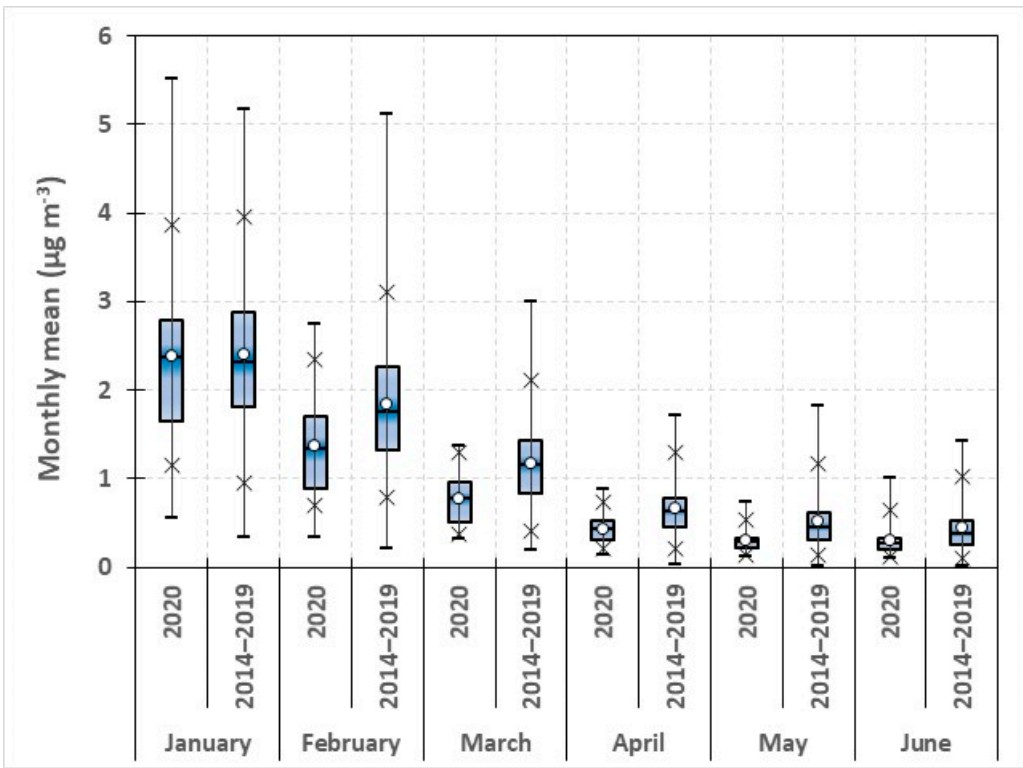

**Figure 8.** Box and whisker plots for the distributions of the monthly mean benzene concentrations in 2020 and 2014–2019. (whiskers: minimum–maximum range; crosses: 5th–95th percentile range; box: q3–q1 range).

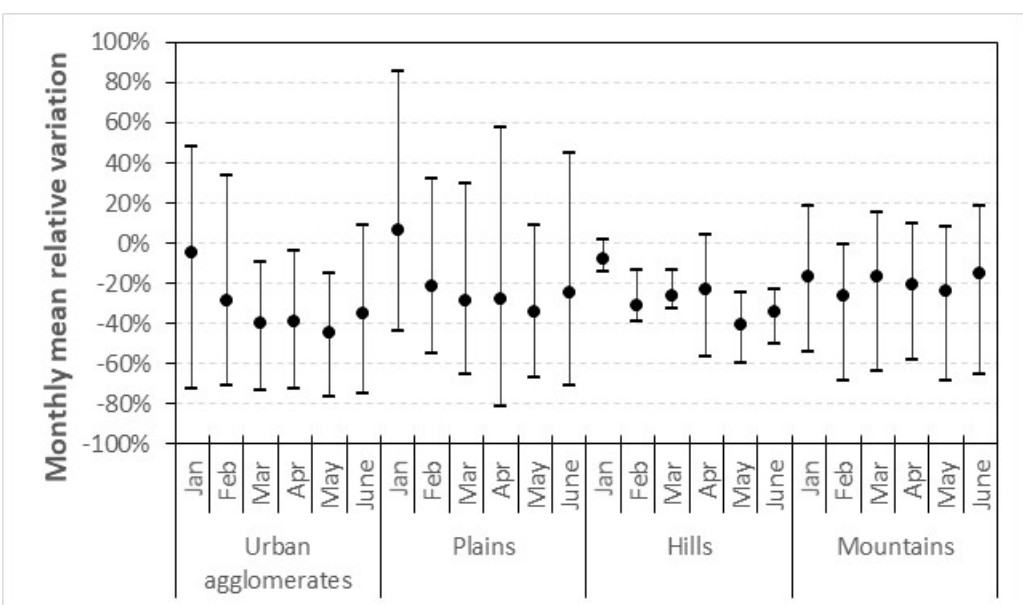

**Figure 9.** Time pattern of the relative variation of benzene monthly concentrations in 2020 with respect to 2014–2019 stratified by station zone (black dot: mean value; bars: minimum–maximum range).

The comparison of time patterns of the overall daily mean and median concentrations in 2020 with the corresponding values of 2014–2019 highlighted once again the progressive reduction of the concentration levels over the whole Po Valley (Figure S7). In January, the overall mean concentrations were usually (65%) within the 2014–2019 range (roughly 1.8–3.0 $\mu g\,m^{-3}$), with a time pattern that mirrored the behavior of $NO_2$. In February, concentrations were still mostly (55%) within the $IQR_{14–19}$, but below $q1_{14–19}$ for the remaining 45%, namely during the low-concentration periods driven by the meteorological conditions already observed for $NO_2$ (e.g., February 4th–5th and 11th–13th) and in the end of February, when the very first local lockdown measures were already enforced in small municipalities of Lombardy. From March on, the overall means were always below $q2_{14–19}$ (<1.5 $\mu g\,m^{-3}$ down to <0.5 $\mu g\,m^{-3}$), and even below $q1_{14–19}$ (<1.0 $\mu g\,m^{-3}$ down to <0.2 $\mu g\,m^{-3}$) for more than 50% of days (Table 5), with the difference between mean values usually being around 0.3 $\mu g\,m^{-3}$, but sometimes as high as 0.5–0.6 $\mu g\,m^{-3}$. The decreasing trend affected the maximum daily values too, almost always lying within the $q3_{14–19}$-$max_{14–19}$ range, down to around 0.7–1.0 $\mu g\,m^{-3}$, in May. In May and June, concentration levels remained fairly stable around 0.3 $\mu g\,m^{-3}$, always below the median values of the previous years (30% lower, roughly), and did not show the slight progressive increase observed for $NO_2$. In general, during the lockdown period the weekly patterns for benzene did not display an evident cycle and were less identifiable than for $NO_2$: indeed, daily concentrations remained basically constant during the whole week, and Sundays' levels did not decrease with respect to weekdays. This pattern suggests that benzene levels are mainly driven by the regional background and less affected by contributions from local emissions. However, stratified analyses showed that a weak weekly pattern could still be recognized at traffic stations in general, and at those located in urban agglomerates in particular, with respect to background stations.

**Table 5.** Frequency distribution (fractional number of days in each month) of the 2020 overall daily mean concentrations by concentration range observed for the 2014–2019 period ($q1_{14-19}$, $q2_{14-19}$, and $q3_{14-19}$: 1st, 2nd, and 3rd quartiles).

| Range | January | February | March | April | May | June |
|---|---|---|---|---|---|---|
| <$q1_{14-19}$ | 16.1% | 44.8% | 77.4% | 53.3% | 61.3% | 10.0% |
| $q2_{14-19}$–$q1_{14-19}$ | 19.4% | 44.8% | 22.6% | 46.7% | 38.7% | 90.0% |
| $q3_{14-19}$–$q2_{14-19}$ | 45.2% | 10.3% | 0.0% | 0.0% | 0.0% | 0.0% |
| >$q3_{14-19}$ | 19.4% | 0.0% | 0.0% | 0.0% | 0.0% | 0.0% |

*3.3. NH₃*

The stations monitoring $NH_3$ are far fewer than those available for monitoring $NO_2$ and benzene, because regulations do not set air quality standards for this pollutant, and thus, its monitoring is not mandatory. In 2020, only 10 stations, mostly located in Lombardia, were in operation out of the 14 active in the six previous years; additionally, one of the stations was not considered in this work, because its location in a very peculiar rural context makes it poorly representative, as discussed by Lonati and Cernuschi [39]. Therefore, data analyses were developed for the whole dataset, without any further stratification.

The comparison between the monthly mean values for 2020 and 2014–2019, both for single station data (Figure 10 left panel) and for the data distributions (Figure 10 right panel), did not show the systematic decrease observed for $NO_2$ and benzene from March on. As summarized in Table 6, concentrations levels were basically higher in January and February 2020 than in the previous years, but lower or similar to the past in the following months. This result is fully consistent with the fact that $NH_3$ almost entirely comes from agriculture, whose emissions were not affected by the restrictive measures of the lockdown. The higher pre-lockdown values (January–February) were likely due to the enhanced slurry spreading activities (land application of liquid manure) that usually take place in autumn, hindered by unfavorable weather conditions in 2019 and thus delayed to the beginning of 2020 [34]. The time trends of the daily concentrations displayed mean and median levels for 2020 fluctuating around the corresponding values of 2014–2019 without any systematic pattern (Figure S8). Given the constant emission regime, this confirmed that no particular meteorological conditions able to affect the ambient concentration levels occurred in 2020. Interestingly, concentration levels at urban stations (in the order of 10–15 µg m$^{-3}$) did not show a decline, even during the month of April, when $NO_2$ and benzene showed the strongest reductions. Thus, the large reduction of $NH_3$ emissions from traffic (estimated at up to −80%) did not affect ambient concentration levels, even at the local scale, seemingly because the atmospheric presence of $NH_3$ in the Po Valley was completely driven by agricultural emission transported all over the area by air masses' local circulation [39].

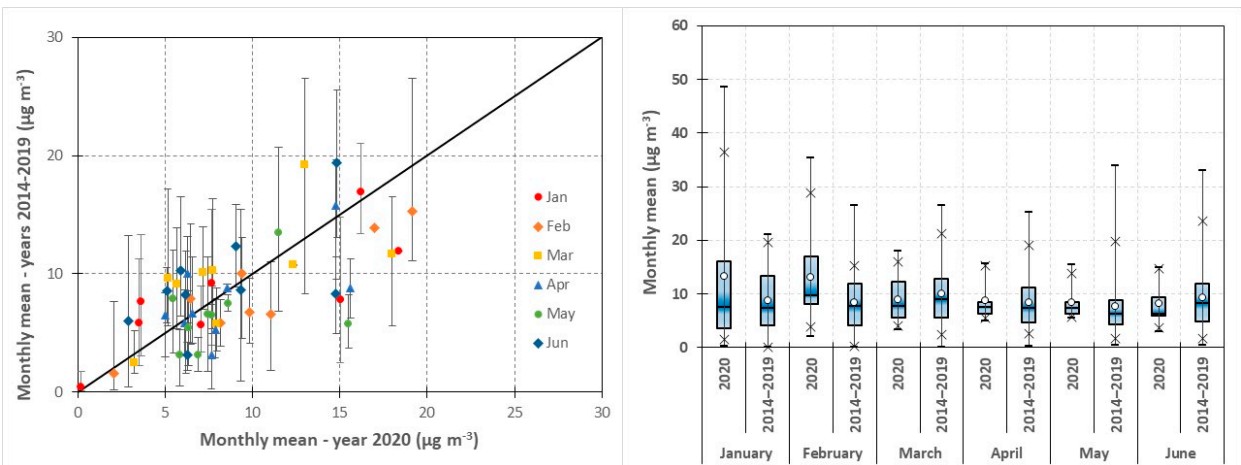

**Figure 10.** Left panel: scatterplot of the monthly mean $NH_3$ concentrations for 2020 versus the corresponding values minimum–maximum range for 2014–2019; right panel: box and whisker plots for the distributions of the monthly mean $NH_3$ concentrations in 2020 and 2014–2019 (whiskers: minimum–maximum range; crosses: 5th–95th percentile range; box: q3–q1 range).

**Table 6.** Summary statistics for $NH_3$ monthly mean concentrations ($\mu g\ m^{-3}$) in 2020 and 2014–2019 (q1, q2, and q3: 1st, 2nd, and 3rd quartiles; p5, p95: 5th, 95th percentiles; N = number of observations).

| Parameter | January | | February | | March | | April | | May | | June | |
|---|---|---|---|---|---|---|---|---|---|---|---|---|
| | 2020 | 2014–19 | 2020 | 2014–19 | 2020 | 2014–19 | 2020 | 2014–19 | 2020 | 2014–19 | 2020 | 2014–19 |
| Mean | 13.4 | 8.7 | 13.2 | 8.4 | 8.9 | 10.0 | 8.7 | 8.4 | 8.4 | 7.7 | 8.3 | 9.4 |
| St. dev. | 14.6 | 5.8 | 9.8 | 5.7 | 4.6 | 5.9 | 3.8 | 5.1 | 3.2 | 6.2 | 4.2 | 6.6 |
| Minimum | 0.2 | 0.1 | 2.1 | 0.1 | 3.3 | 0.1 | 5.0 | 0.3 | 5.5 | 0.5 | 2.9 | 0.5 |
| Maximum | 48.5 | 21.0 | 35.4 | 26.6 | 18.0 | 26.6 | 15.6 | 25.3 | 15.5 | 33.9 | 14.8 | 33.0 |
| q1 | 3.6 | 4.2 | 8.2 | 4.2 | 5.7 | 5.6 | 6.3 | 4.8 | 6.3 | 4.4 | 5.9 | 4.9 |
| q2 | 7.7 | 7.4 | 9.9 | 7.8 | 7.7 | 9.0 | 7.6 | 7.3 | 7.4 | 6.3 | 6.3 | 8.4 |
| q3 | 16.2 | 13.4 | 17.0 | 12.0 | 12.3 | 12.9 | 8.5 | 11.3 | 8.6 | 8.9 | 9.3 | 11.9 |
| p5 | 1.5 | 0.1 | 3.8 | 0.3 | 4.0 | 2.4 | 5.4 | 2.6 | 5.6 | 1.7 | 3.8 | 1.6 |
| p95 | 36.5 | 19.6 | 28.9 | 15.4 | 16.0 | 21.3 | 15.3 | 19.1 | 13.9 | 19.9 | 14.8 | 23.6 |
| N | 9 | 48 | 9 | 48 | 9 | 52 | 9 | 48 | 9 | 45 | 9 | 45 |
| Means Test | non reject | | non reject | | non reject | | non reject | | non reject | | non reject | |

## 4. Conclusions

As in many areas worldwide where restrictions to human activities were implemented in order to limit the spread of the COVID-19 virus, the Po Valley also experienced better air quality during spring 2020. However, as far as the gaseous pollutants considered in this work are concerned ($NO_2$, benzene, and $NH_3$) air quality improvement was seemingly affected by traffic-related pollutants only. Lockdown rules in Northern Italy determined reductions of nitrogen oxide and benzene emissions from road traffic in the order of −35% of the regional average, but with much higher reductions at traffic sites in urban areas during some weeks in April. Consequently, significant average reductions in the −40% to −35% range were observed for 2020 monthly mean concentrations for the area compared with the 2014–2019 period. In contrast, $NH_3$ ambient concentration levels, almost entirely due to emissions from the agricultural sector, did not experience any relevant changes. Even at traffic stations in urban areas, the role of $NH_3$ traffic emissions was still marginal, or practically negligible, with respect to agriculture, whose emissions affected $NH_3$ levels over the entire air basin of the Po Valley.

Additional pieces of information highlighted by this work can be summarized as follows:

- The long-term air quality limit for $NO_2$ (40 μg m$^{-3}$ as annual average) is likely to be respected at all the monitoring stations of the Po Valley in 2020, due to the low concentration levels recorded from March to June.
- The observed reductions for the concentration levels were consistent with what could be expected based on emission inventory and source activity data: this supports the accuracy of both these factors, and thus, the reliability of the emissions scenario during the lockdown period to be used for testing the performance of air quality models at the regional scale.
- The Po Valley appears as a rather homogeneous air basin, with urban area hot-spots where the contributions of the local emissions add up to a relatively high regional background concentration level. Indeed, the low regional background reached at the end of the lockdown period was beneficial for the following period, namely, with concentration levels in June 2020 still below the average of the previous years, in spite of the resumption of pre-lockdown activities.
- The relatively slow response of the air quality levels to the sudden decrease of the emissions confirms that the Po Valley is an air basin with a weak exchange of air masses, which favors both the build-up of atmospheric pollutants and the development of secondary formation processes.

The improvement in air quality for traffic-related pollutants observed in the Po Valley during and after the 2020 spring lockdown was the result of exceptional measures enforced over the whole region for a prolonged time period. Both those features (i.e., territorial extent and temporal continuity) are typical of structural interventions for air quality management at the central level, in opposition to local-level policies, usually limited to urban areas and mostly enforced for short time periods due to critical pollution events. Thus, national policies for air quality should aim at a baseline limitation of traffic emissions and be coupled with further coordinated policies, both structural and emergency, by the regional authorities of the Po Valley that consider the abovementioned peculiar features of the area. These policies can directly address the traffic source (i.e., supporting the renewal of the circulating fleet, public transportation systems, and intermodal freight transport) but can also address the individual mobility demand. Actually, the changes in habits and working modes experienced during the lockdown period have demonstrated that teleworking and remote business meetings, previously scarcely used, can profitably reduce the need for mobility, and thus help reduce atmospheric emissions from road traffic with beneficial outcomes for the air quality.

**Supplementary Materials:** The following are available online at https://www.mdpi.com/2073-4433/12/2/264/s1, Figure S1: Relative variation of NOx (top), NMVOC (middle), $NH_3$ (bottom) emission by zones in Lombardia. [32]; Figure S2: Pie-charts for the distribution of the relative variation of the monthly mean $NO_2$ concentrations between 2020 and 2014–2019; Figure S3: Weekly time patterns of $NO_2$ overall daily mean concentrations for 2020 versus the corresponding values for 2014–2019. Shaded areas show concentration ranges for 2014–2019; Figure S4: Daily time patterns of $NO_2$ overall hourly mean concentrations for 2020 versus the corresponding values and min-max range for 2014–2019; Figure S5: Scatterplots of the monthly mean benzene concentrations for 2020 versus the corresponding values for 2014–2019. Error bars show the min-max range for 2014–2019. Please note that concentration scales are different for every other two months; Figure S6: Pie-charts for the distribution of the relative variation of the monthly mean benzene concentrations between 2020 and 2014–2019; Figure S7: Time pattern of benzene overall daily mean concentrations for 2020 versus the corresponding values for 2014–2019. Shaded areas show concentration ranges for 2014–2019; Figure S8: Time pattern of $NH_3$ overall daily mean concentrations for 2020 versus the corresponding values for 2014–2019. Shaded areas show concentration ranges for 2014–2019.

**Author Contributions:** Conceptualization and methodology, G.L.; formal analysis, G.L.; data curation and investigation, F.R.; writing—original draft preparation, G.L.; writing—review and editing, G.L. and F.R. All authors have read and agreed to the published version of the manuscript.

**Funding:** This research received no external funding.

**Institutional Review Board Statement:** Not applicable.

**Informed Consent Statement:** Not applicable.

**Data Availability Statement:** The data presented in this study are available on request from the corresponding author.

**Acknowledgments:** The authors gratefully thank the Regional Environmental Agencies (ARPA) of Emilia-Romagna, Lombardia, Piemonte, and Veneto region for supporting access to 2020 data.

**Conflicts of Interest:** The authors declare no conflict of interest.

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
