# Peer review of "Regional Scale Impact of the COVID-19 Lockdown on Air Quality: Gaseous Pollutants in the Po Valley, Northern Italy"

_atmosphere, doi:10.3390/atmos12020264_

Round 1

Reviewer 1 Report

This is an interesting study examining the impact of the COVID-19 lockdown on air quality at a regional scale, i.e. investigating changes in the concentrations of 3 gaseous pollutants in the Po valley, in Northern Italy. It merits publication in Atmosphere once a number of issues are addressed. More specifically:

Main comments

  1. Some improvement in the use of English language is needed. The manuscript should be proofread in this respect.

  1. Abstract: The last sentence is very long and difficult to understand. It should be broken down to smaller sentences, highlighting more clearly the main conclusions (and maybe also providing some policy suggestions).

  1. Introduction: It is not clear where the information presented in Figure 1 comes from. Is it all from ISPRA (2016)? Reference should be provided.

  1. Page 3, lines 122-124 (“Data processing considered the January-June period so that pre-, during-, and post-lockdown conditions of air quality could be separately analyzed.”): However, due to this period covering 6 months, some seasonal effects could be also present, between winter and spring months (regarding emission patterns and meteorological conditions). The authors should provide further insight to this here and link it to the consecutive analysis (e.g. Section 2.1 and Table 2, discussion of Table 3, etc.).

  1. Table 3 and Table 5: It would be useful if the legend of Tables 3 and 5 explains the q-ranges (as in Tables 2 and 4).

  1. Page 9, line 268: Do the authors mean q3-q2? (instead of q3-q1). Also, the meaning of IQR should be better explained in “Materials and methods”.

  1. Page 12, lines 328-330 (“Such a weekend effect becomes stronger in the lockdown period, because traffic reached its minimum levels”): This is not very clear. One would expect that the concentration difference between weekdays and Sundays is larger before the lockdown, when there was full traffic during working days. The authors need to further explain and provide some insight.

  1. Page 15, lines 412-414: Again, the weekly pattern for benzene should be further explained, as well as its differences from the weekly pattern for NO2 (see comment 7 above).

  1. The man question that rises in relation to this paper is what its added value is in relation to other relevant papers on pollutant concentration changes during the Covid-19 lockdown period (especially since such papers also examine areas/cities of Northern Italy). The authors provide some explanation in the beginning, but this needs to be further substantiated in the discussion and conclusions of the article. I believe that this needs to be done in two distinct ways:

(a) The authors should provide further insight on the causality of the results/concentration levels and especially on the differences between pollutants (mainly NO2 and benzene, since ammonia is attributed to agriculture)

(b) the authors need to further discuss on the policy implications and provide suggestions. Especially since policies are usually taken at the central level and their work examines a large area of the country with different types of monitoring stations and pollutant sources (and not only one city). This is only partially done in the last bullet of the conclusions currently.

  1. Supplementary material: If the Supplementary material is to be published/available with the article, all captions and notes should be in English (especially relevant for Figure S1).

Reviewer 2 Report

This is a valuable descriptive study of the Po valley air quality changes due to the natural experiment created by the COVID-19 restrictions in economic activities. I recommend the acceptance of this paper for publication after the following edits have been made.

It will be helpful if the authors will clearly state the objectives at the end of the introduction.

I recommend presenting Figure 2 in landscape orientation on a full page, so the detail is more readily visible. It would also be helpful to see which stations report which pollutants; therefore the spatial distribution and potential biases will be more apparent.

Please write ALL descriptions and results in past tense because the emissions data are from the past.

Note that ‘i.e.’ and ‘e.g.’ do not need a colon ‘:’ after them – please remove all instances

My recommendations and comments are preceded by the Page-Line numbers:

1-6 Use proper subscript for NO2 (and throughout the abstract)

2-50 Use proper superscript for km2

2-62 Change ‘sensitively’ to ‘especially’

2-69 Change ‘the usual’ to ‘typical’

2-73 Change ‘one case of positivity’ with ‘one positive case’

2-81 Remove colon ‘:’ after ‘i.e.’

2-82 Change ‘the most part of all’ to ‘most’

2-88 Entire paragraph should be written in the past tense

2-96 Change ‘of 47% as an average when considering medium cities too’ to ‘for an average of 47% in medium sized cities’

3-101 Define the ‘SNAP’ acronym in the Figure 1 caption

3-103 Change ‘experienced different variations’ to ‘varied’ [emissions don’t experience anything, but they do vary, which means their values are different, so simplify this awkward statement]

3-106 Change ‘indicate’ to ‘indicated’ – write all descriptions in past tense

3-109 Remove ‘even in excess of’ and remove ‘down to’ [simplify the wordiness for clarity]

3-115 Change ‘due to its almost total origin’ to ‘originating primarily’

3-125 Change ‘without and’ to ‘without any’

3-127 Insert ‘wasn’t’ before ‘able’ and remove colon ‘:’ after ‘i.e.’

3-132 It would be extremely helpful and is typically expected to see your study objective clearly stated at the end of the introduction section

4-135 Insert comma (,) after ‘In the Po valley’

4-137 Insert ‘the’ before ‘European’

4-147 Legend font size and scale bar must be increased in both map panels

4-154 Remove colon ‘:’ after ‘e.g.’

5-157 Change ‘kind’ to ‘type’

5-158 Remove colon ‘:’ after ‘i.e.’

5-159 Change ‘all the sources upwind’ to ‘all upwind sources’

5-160 Insert ‘the’ before ‘regional’

5-161 Remove colon ‘:’ after ‘i.e.’

5-173 Change ‘kind’ to ‘type’

5-178 Change ‘Number of data’ to ‘Number of observations’

5-181 Change ‘have been’ to ‘were’ [do this for all instances of ‘have been’ in the remaining text]

5-184 Change ‘consequently to’ to ‘as a consequence of the’

6-185 Change ‘those’ to ‘the mean of’

6-186 Change ‘resolution’ to resolutions’; change ‘for’ to ‘to’

6-188 Change ‘level’ to ‘levels’; change ‘thanks to’ to ‘due to’

6-191 Insert ‘to’ after ‘according’

6-193 Remove colon ‘:’ after ‘i.e.’

6-198 Change ‘level’ to ‘levels’

6-204 Change ‘dataset’ to ‘datasets’

6-206 Change ‘Table 3.1’ to ‘Table 3’

7-227 Use same x- and y-axis intervals for all months – May and June only go to 80, while Jan and Feb go to 120 and March and April go to 100

8-234 Indicate NO2 in the Table 3 caption

8-246 Change ‘kind’ to ‘type’

8-250 Change ‘kind’ to ‘type’

9-269 Remove colon ‘:’ after ‘e.g.’

10-287 Use different line symbols for Min 2020 and Max 2020 [there is the occasional stupid/lazy reader out there]

10-288 Change ‘Colored’ to ‘shaded’

10-289 The meaning of ‘acquisition obtained’ is unclear [I am not sure what to suggest here]

11-301 Change ‘further-more’ to ‘further’

11-312 Use different line symbols for Min 2020 and Max 2020 [there is the occasional stupid/lazy reader out there]; adjust the graph titles so that the panels align - perhaps locate ‘Traffic stations’ and Background stations’ on separate lines after ‘Urban agglomerates’

11-313 Change ‘Colored’ to ‘shaded’

12-316 Change ‘for’ to ‘of’; change ‘kind’ to ‘type’

12-318 Remove colon ‘:’ after ‘e.g.’

12-319 Change ‘that’ to ‘than’

12-323 How does the reader know which calendar day corresponds to Sundays?

12-338 Insert ‘an’ before ‘hourly’

13-377 Remove ‘kind’

14-390 Change ‘Number of data’ to ‘Number of observations’; use consistent lower or upper case for the text in the ‘Mean Test. And ‘K-S test’ rows [why is the text for the May column capitalized?]

15-407 Change ‘concentrations’ to ‘concentration’

15-409 Remove ‘, conversely’

15-410 Change ‘is less as for’ to ‘and is less than’

15-419 Table 1 shows 14 total stations for NH3, and here you state 10 [are you describing only Lombardia in this section?]

15-426 Change ‘than’ to ‘and’

15-432 [you may wish to define ‘slurry spreading’ for the uninformed reader]

16-450 Change ‘Number of data’ to ‘Number of observations’

16-454 Change ‘also the Po valley’ to ‘the Po valley also’

16-455 Remove ‘a’

16-456 Change ‘improvement affected’ to ‘improvement was affected by’

16-459 Change ‘as’ to ‘of the’

17-469 Change ‘whose’ to ‘where’

17-470 What do you meant by ‘respected’ all over the region?

17-471 Change ‘thanks to ‘ to ‘due to’; change ‘form’ to ‘from’

17-477 Change ‘areas’ to ‘area’

17-489 Change ‘reducing’ to ‘reduce’

Round 2

Reviewer 1 Report

The authors adequately responded to my comments on the original manuscript.

Author Response

We thank the reviewer for the valuable comments and suggestions that helped improving our original manuscript.